# The Specific ROCK2 Inhibitor KD025 Alleviates Glycolysis through Modulating STAT3-, CSTA- and S1PR3-Linked Signaling in Human Trabecular Meshwork Cells

**DOI:** 10.3390/biomedicines12061165

**Published:** 2024-05-24

**Authors:** Megumi Watanabe, Tatsuya Sato, Araya Umetsu, Toshifumi Ogawa, Nami Nishikiori, Megumi Suzuki, Masato Furuhashi, Hiroshi Ohguro

**Affiliations:** 1Departments of Ophthalmology, School of Medicine, Sapporo Medical University, S1W17, Chuo-ku, Sapporo 060-8556, Japan; watanabe@sapmed.ac.jp (M.W.); araya.umetsu@sapmed.ac.jp (A.U.); nami076@yahoo.co.jp (N.N.); megumi.h@sapmed.ac.jp (M.S.); 2Departments of Cardiovascular, Renal and Metabolic Medicine, Sapporo Medical University, S1W17, Chuo-ku, Sapporo 060-8556, Japan; satatsu.bear@gmail.com (T.S.); a08m024@yahoo.co.jp (T.O.); furuhasi@sapmed.ac.jp (M.F.); 3Departments of Cellular Physiology and Signal Transduction, Sapporo Medical University, S1W17, Chuo-ku, Sapporo 060-8556, Japan

**Keywords:** XF real-time ATP rate assay, RNA sequencing, KD025, Rho-associated coiled-coil containing protein kinase (ROCK), human trabecular meshwork (HTM)

## Abstract

To investigate the biological significance of Rho-associated coiled-coil-containing protein kinase (ROCK) 2 in the human trabecular meshwork (HTM), changes in both metabolic phenotype and gene expression patterns against a specific ROCK2 inhibitor KD025 were assessed in planar-cultured HTM cells. A seahorse real-time ATP rate assay revealed that administration of KD025 significantly suppressed glycolytic ATP production rate and increased mitochondrial ATP production rate in HTM cells. RNA sequencing analysis revealed that 380 down-regulated and 602 up-regulated differentially expressed genes (DEGs) were identified in HTM cells treated with KD025 compared with those that were untreated. Gene ontology analysis revealed that DEGs were more frequently related to the plasma membrane, extracellular components and integral cellular components among cellular components, and related to signaling receptor binding and activity and protein heterodimerization activity among molecular functions. Ingenuity Pathway Analysis (IPA) revealed that the detected DEGs were associated with basic cellular biological and physiological properties, including cellular movement, development, growth, proliferation, signaling and interaction, all of which are associated with cellular metabolism. Furthermore, the upstream regulator analysis and causal network analysis estimated IL-6, STAT3, CSTA and S1PR3 as possible regulators. Current findings herein indicate that ROCK2 mediates the IL-6/STAT3-, CSTA- and S1PR3-linked signaling related to basic biological activities such as glycolysis in HTM cells.

## 1. Introduction

Among the families of the serine-threonine protein kinase family, rho-associated coiled-coil-containing protein kinases, ROCKs are known as pivotal factors regulating remodeling of the actin cytoskeleton [1,2,3,4,5] and they function to regulate cell architecture/adhesion/motility, cell differentiation, cell apoptosis, and inflammation [6,7,8]. Two isoforms of ROCKs, ROCK1 and ROCK2, having homologous amino acid compositions of the C-termini, catalytic kinase domains and Rho-binding domains and distinct the coiled-coil regions are identified [9,10]. Despite having homologous compositions, ROCK1 and 2 may have diverse functions and distinct downstream targets. For instance, ROCK2, but not ROCK1, phosphorylates Ser19 residue of myosin light chain (MLC), that is also phosphorylated by MLC kinase (MLCK), resulting altering the Ca^2+^ sensitive skeletal muscle cell contraction [11].

Within ocular and peri-ocular tissues including the trabecular meshwork (TM), ciliary muscles, and retina, ROCKs are also expressed [9,10] and function in the ocular physiology and pathology of various ocular diseases including corneal dysfunction, cataract genesis and retinopathy [1,2,12,13,14,15]. Therefore, ROCKs may be promising therapeutic candidates for these ocular diseases, especially glaucoma. In fact, it has been shown that expression of both ROCK1 and ROCK2 are identified in human TM (HTM) cells [16] and that ROCK inhibitors (ROCK-is) substantially decrease intraocular pressure (IOP) in various animal models [17,18]. In Japan, ripasudil hydrochloride hydrate (Rip), a pan-ROCK-is is already used as a hypotensive therapy for patients with glaucoma as well as ocular hypertension [19,20]. As an underlying mechanism causing hypotensive activity by Rip, it was shown that Rip enhances outflow facility of humor (AH) by modulating the permeability at HTM in association with their disruption of tight junctions of HTM cells [21]. To support this mechanism, we independently developed an in vitro three-dimensionally (3D) cultured HTM model replicating the multi-layer sheets architecture of the HTM in addition to an in vitro model of an HTM monolayer by a two-dimensionally (2D) cell planar culture. Upon administering transforming growth factor-β2 (TGF-β2) or dexamethasone (DEX) to 2D and 3D cultured HTM cells, we observed the following effects; (1) significantly downsized and harder 3D HTM spheroids, (2) a substantial increase in the trans-epithelial electrical resistance (TEER) values, and (3) those changes were different between TGF-β2 and DEX treatments. Based on these results, we suggested that those models may become two different in vitro glaucomatous HTM models, that is, an in vitro primary open angle glaucoma (POAG) HTM model and a steroid-induced glaucoma (SG) HTM model, respectively [22]. Using these models, we found that the effects of TGF-β2 or DEX were substantially decreased by administration of pan-ROCK inhibitors, Rip and Y27632 [23], suggesting that pan-ROCK-is indeed have beneficial effects on AH outflow facility at TM levels. However, the efficacies of Rip and Y27632 were different, in which their inhibitory activities for ROCK1 or ROCK2 are diverse, that is, ROCK 1 inhibition is lower and higher that ROCK2 inhibition in Rip and Y-27632, respectively [24,25], suggesting that ROCK1 and ROCK2 diversely regulate the biological activities of the HTM. In fact, we also found that the effects of a specific ROCK2 inhibitor, KD025, on the physical aspects of glaucomatous HTM cell model treated by TGF-β2 or DEX was quite different from the effects of Rip [26,27]. However, as of this writing, there is limited information on the biological regulations of HTM physiology by ROCK2.

Therefore, to study the biological roles of ROCK2 in physiological properties of the HTM, we investigated functional effects of KD025 on planar-cultured HTM by performing real-time ATP rate assay using a Seahorse XFe analyzer (Agilent Technologies, Santa Clara, CA, USA). Furthermore, we performed RNA sequencing analysis to elucidate the link between altered metabolic functions and gene expression patterns in HTM cells treated or not treated with the selective ROCK2 inhibitor KD025.

## 2. Materials and Methods

### 2.1. Human Trabecular Meshwork (HTM) Cells

All experimental procedures using human-derived cells were taken place in compliance with the tenets of the Declaration of Helsinki and approved by the internal review board of Sapporo Medical University. Commercially available human trabecular meshwork (HTM, Applied Biological Materials Inc., Richmond, BC, Canada) was used in the present study. In advance, those HTM cells were confirmed to ensure truly TM cells as described by Keller et al. [28]

### 2.2. Real-Time ATP Rate Assay in Planar 2D Cultured HTM Cells by Using a Seahorse XF96e Analyzer

The 20,000 2D cultured HTM cells were placed in each well of an XFe96 Cell Culture Microplate (Agilent Technologies, #103794-100, Santa Clara, CA, USA) approximately 24 h before the assay, and the plate was incubated under standard humid and normoxia conditions (37 °C, 5% CO_2_). Assay buffer containing dimethyl sulfoxide (DMSO, control) or ROCK2 inhibitor KD025 (final concentration: 10 µM) was subjected to measurements of oxygen consumption rate (OCR) and extracellular acidification rate (ECAR) in the cells were simultaneously evaluated using a Seahorse XFe96 Bioanalyzer (Agilent Technologies, Santa Clara, CA, USA) as described previously [29,30]. The glycolytic and mitochondrial ATP production rates after adding DMSO or KD025 were calculated using Seahorse Analytics 1.0.0.-520 (https://seahorseanalytics.agilent.com).

### 2.3. RNA Sequencing Analysis and IPA Gene Function and Pathway Analysis

Total RNA isolated from 2D confluent HTM cells untreated or treated with 10 μM KD025 for 24 h in a 150 mm dish as described above (*n* = 3) using an RNeasy mini kit (Qiagen, Valencia, CA, USA) according to the manufacturer’s instructions. Followingly, RNA extraction and next-generation sequencing were performed as described recently [31]. Obtained sequence data were filtered using FastQC software (version 0.11.7), checked their quality control by an Agilent 2100 Bioanalyzer (Agilent, CA, USA) and Trimmomatic (version 0.38) and mapped to the reference genome sequence (GRCh38) using HISAT2-2.1.1 tools software [32]. The read counting for each respective gene and statistical analysis were processed using featureCounts (version 1.6.3) and DESeq2 (version 1.24.0), respectively. Differentially expressed genes (DEGs) were determined as genes with fold-change ≥2.0 and false discovery rate (FDR)-adjusted *p*-value  <  0.05 and *q*  <  0.08 between groups.

Ingenuity pathway analysis (IPA, Qiagen, https://digitalinsights.qiagen.com/products-overview/discovery-insights-portfolio/analysis-and-visualization/qiagen-ipa/, the accessed date 9 February 2024) [33] was used for further analysis to predict various pathways by uploading the significant up-regulated and down-regulated DEGs excel file to IPA core analyses. Enrichment of the particular genes in networks in IPA was evaluated using Fisher’s exact test. In addition, the IPA software orders top functions related with each network based on the enrichment scores (z-score) and predicts possible upstream regulator and causal network regulator as shown in recent studies [33,34,35].

### 2.4. Other Analytical Methods

Real-time PCR were carried out essentially as previously reported [36,37] using predesigned primers (Appendix A). The expression of each respective gene was normalized by using the expression of a housekeeping gene 36B4 (Rplp0). As experimental data, the arithmetic mean ± the standard error of the mean (SEM) was used in conjugation with statistical analyses essentially as described in our previous reports [36,37].

## 3. Results

To identify the biological roles of ROCK2 in conventional AH outflow, especially at the TM level, initially we evaluated effects of specific ROCK2 inhibitor KD025 on cellular metabolic functions by assessing both glycolytic and mitochondrial ATP production rates in planar-cultured HTM cells. Addition of KD025 on HTM cells induced an increase in OCR values (Figure 1A) and a dramatical reduction in ECAR values (Figure 1B). Especially, compensatory increase in ECAR values by adding ATP synthase inhibitor oligomycin was abolished in the presence of KD025 (Figure 1B). Consistent with the overall results of OCR and ECAR values, the glycolytic ATP production rate after adding KD025 (29.5 ± 4.1 pmol/min/μg protein) was significantly and dramatically lower than that after adding DMSO (63.5 ± 6.5 pmol/min/μg protein) (Figure 1C). Conversely, the mitochondrial ATP production rate after adding KD025 (71.1 ± 11.0 pmol/min/μg protein) was significantly higher than that after adding DMSO (49.9 ± 5.8 pmol/min/μg protein) although there was a statistical difference in total ATP production rates between two groups (Figure 1C). These results suggest that KD025 has an effect in shifting the metabolism to produce ATP from glycolysis to mitochondrial respiration in planar-cultured HTM cells.

RNA sequencing analysis was performed using HTM cells not treated (NT, *n* = 3) and treated with KD025 (KD, *n* = 3). As shown in a heatmap (Figure 2) and in an M-A (Figure 3A) and a volcano plot (Figure 3B), there were 602 substantially up-regulated and 380 substantially down-regulated genes termed differentially expressed genes (DEGs) in NT cells and KD cells with a significance level of false discovery rate (FDR) less than 0.05 and an absolute fold-change more than 2 was identified (a list of all of the up-regulated or down-regulated DEGs is included in a Appendix A). As the most prominent DEGs, the top 10 of up-regulated and down-regulated DEGs are indicated in Table 1.

GO ontology analysis was performed to estimate unidentified biological effects of the specific ROCK 2 inhibitor KD025 on HTM cells. The detected DEGs were more abundantly categorized in GO terms related to plasma membrane, extracellular component and integral cellular component among the cellular components (Figure 4A) and those related to signaling receptor binding and activity and protein heterodimerization activity among the molecular functions (Figure 4B).

Next, we conducted Ingenuity Pathway Analysis (IPA) to estimate that what kinds of biological functions and networks are underlaid in the KD025-induced effects on HTM cells based upon 602 significantly up-regulated and 380 significantly down-regulated DEGs. The estimated molecular and cellular functions of the detected DEGs were related to basic cellular biological properties including cellular movement, development, growth, proliferation, signaling and interaction (Table 2), and they were associated with various diseases networks in addition to cell death and survival (Table 3). These results were quite rational because ROCKs mediate lots of pivotal cellular functions such as cell architecture, motility, secretion, proliferation, and gene expression, and those pathways are also related to other signaling pathways contributing to the pathogenesis of various systemic diseases [38].

To study further, we estimated what kinds of upstream regulatory mechanisms were involved. IPA analysis suggested various candidates of molecules and factors as the possible upstream and causal network regulators. However, most of those were chemical drugs, chemical reagents, chemical toxicants, peptides in addition to a few cytokines, growth factors, kinases and transcription regulators. Among the latter molecules, interleukin 6 (IL6) and signal transducer and activator of transcription 3 (STAT3), and cystatin A (CSTA) and sphingosine-1-phosphate (S1P) receptor 3 (S1PR3) were determined as possible upstream regulators and a possible causal network regulator, respectively, because changes of expression rates of those four factors (expression log ratio more than |1.5|, IL6; −1.688, STAT3; 1.51, CSTA; 5.41 and S1PR3; 1.52) were much larger than those of others (Table 4). As shown in Figure 5, upon administering 10 μM KD025, levels of gene expression of IL6 was moderately increased and those of other three genes were significantly decreased.

## 4. Discussion

It is shown that ROCK1 and ROCK2 are ubiquitously expressed throughout embryogenesis and in adult tissues [9,39]. Although both ROCKs are involved in the similar roles including cell adhesion and microfilament assembly, they are also differently involved in actin-mediated extracellular matrix formation [40,41] and cell adhesion to fibronectin [42,43]. Furthermore, both ROCKs are differently localized within various tissues and organs [44]. For instance, in thymus and blood, there is predominant ROCK1 expression and little or no ROCK2 expression despite highly expressed ROCK2 in the eyes and in cardiac and brain tissues [9,45,46]. Collectively, we assume that hypotensive effects of ROCK-is on IOP could primarily be induced by ROCK2 inhibition rather than ROCK1 inhibition. However, at present, insufficient information on ROCK1 has been available because specific ROCK1 inhibitors are present. In our preceding study, to elucidate which ROCK1 or ROCK2 inhibition of ROCK-is contributes on their hypotensive efficacy, we compared the effects of a pan-ROCK-i, Rip and KD025 on various physical and biochemical properties of in vitro glaucomatous HTM models using 2D and 3D cultured HTM cells [22] supplemented without or with TGF-β2 [23,26] or DEX [27]. As shown in Table 5, Rip and KD025 had significantly different effects in both TGF-β2 and DEX-treated models on 1) physical properties including TEER of 2D HTM monolayers and 3D HTM spheroids’ size and stiffness and 2) cellular metabolic functions related to mitochondria and glycolysis of 2D cultured HTM cells. Interestingly, these physical and cellular metabolic properties were greatly different between treatments with Rip and KD025, and even opposite effects of the two drugs on some of these properties, especially physical properties of 3D HTM spheroids, were observed. Here, IPA analysis suggested that STAT3 and IL6 are possible upstream regulators for KD025-induced DEGs. Interestingly, since STAT3- and IL6-related signaling are known to be pivotally involved in the mitochondrial function [47] and glycolysis [48], KD025-induced metabolic shift to produce ATP from glycolysis to mitochondrial respiration in HTM cells (Figure 1) was quite rationally understood. Furthermore, in our previous analysis of RNA sequencing of 2D and 3D cultured 3T3-L1 cells, STAT3 and IL6 were also estimated as the master upstream and causal network regulators for inducing 3D spheroid formation [49]. Therefore, taken together, if KD025 effects were indeed regulated by the STAT3- and IL6-related signaling axis, we rationally understood that the effects of KD025 in the in vitro glaucomatous HTM models were more apparent in the 3D spheroid model than in the 2D model.

It has been shown that IL6 signaling is not only involved in cancer, fibrosis, and chronic inflammation [50,51,52], but may also be related to glaucoma by regulation of IOP [53,54,55]. Classic and trans-signaling axes have been identified as the main signaling of IL-6 [56]. In the classic signaling, IL6 forms an IL6–mIL6R complex by binding to its membrane-bound receptor, mIL6R, and thereafter binds with the receptor subunit glycoprotein 130 (gp130) activating the canonical Janus Kinase (JAK)/ STAT or noncanonical mitogen-activated protein kinase (MAPK) [56,57]. In support of this finding, an increased effect of IL-6 on AH outflow facility was observed in perfused ex vivo organ cultures of porcine anterior segment [58]. On the other hand, in trans-signaling, IL6 binds to the soluble form of mIL6R (sIL6R), resulting unfavorable inflammation by binding binds with gp130 [56,57]. In fact, it was shown that the levels of sIL6R were increased in the AH of patients with POAG as compared with those in age-matched controls, [53] suggesting that IL6 trans-signaling may intricately affect AH homeostasis [58]. As of this writing, the underlying mechanisms of IL6-induced effects on AH homeostasis of HTM, especially glaucomatous phenotype models, have not been elucidated yet. However, a previous study showed that IL-6–mediated trans-signaling potently suppresses TGF-β signaling in HTM cells [53], and another study using an HTM cell model revealed a synergistic cross-linkage between lysophosphatidic acid (LPA) and IL6 trans-signaling mechanisms related to YAP, TAZ, and pSTAT3 [59]. It was also shown that ROCK2 signaling causes activation of IKKβ/NF-κB/IL-6/STAT3 by a positive feedback manner that results in resistance of anti-tumor drugs in hepatocellular carcinoma (HCC) cells [60]. Collectively, these observations strongly support the results of our study showing that IL6 and STAT3 are possible upstream regulators for ROCK2 activity of HTM cells.

In addition to IL6 and STAT3, sphingosine-1-phosphate (S1P) receptor 3 (S1PR3) was estimated as a possible causal network regulator. It was shown that S1P caused an increase in resistance of the AH outflow of both human and porcine eyes [61,62] by binding to S1P receptors [63]. Five subtypes of SIP receptors (S1PR1–5) have been identified, and it has been shown that S1PR1–3 are ubiquitously expressed, whereas S1PR4 is expressed only in lymphoid tissue and SIPR5 is expressed only in brain and skin tissues [64]. In TM cell culture models, S1P activates Rho, thereby facilitating phosphorylation of myosin light chain (MLC) and increasing stress fiber formation [61]. However, another study showed that activation of S1PR2, but not that of S1PR1 or S1PR3, increases conventional outflow resistance to regulate IOP [65]. Furthermore, a member of the cystatin superfamily of proteins, cystatin A (CSTA) that activates cysteine protease inhibitors [66] was identified as significant up-regulated DEG. Interestingly, a previous study indicated that CSTA could reduce cleavage of wild-type MYOC in primary TM cells. However, such CSTA-induced effects were not observed in an inactively mutated CSTA, suggesting that CSTA may become a potential biomarker and therapeutic target for MYOC-induced glaucoma [67]. However, as limitations of the present study, the following issues remain to be elucidated; (1) why ROCK2 inhibition by KD025 caused diverse effects in different types of in vitro glaucomatous HTM models, (2) roles of ROCK1 actions in HTM cells and (3) correlations of both S1PR3 and CSTA with ROCK2 activities in HTM cells. Furthermore, since KD025 is a selective, ATP-competitive inhibitor of human ROCK2 (IC50 value of 105 nM) with some effects on human ROCK1 (IC50 value of 24 µM) [68], some little inhibitory effects against ROCK1 (approximately 20–30%) in contrast to 90% inhibitory effects against ROCK2 may be non-negligible in the case of present experimental condition of 10 μM KD025. Therefore, additional investigations using specific inhibitors for downstream factors related to STAT3/IL6 and other candidates as well as another ROCK2-specific inhibitors will be carried out as our next research projects.

In conclusion, we found that a specific ROCK2 inhibitor, KD025, significantly suppressed glycolytic ATP production rate and increased mitochondrial ATP production rate in HTM cells. In addition, RNA sequencing analysis estimated that 602 up-regulated and 380 down-regulated DEGs between KD025 untreated and treated HTM cells. Based on IPA analysis of these DEGs, it was estimated that ROCK2 inhibition by KD025 modulated IL-6/STAT3-, CSTA- and S1PR3-linked signaling related to basic biological activities such as glycolysis in HTM cells.

## Figures and Tables

**Figure 1 biomedicines-12-01165-f001:**
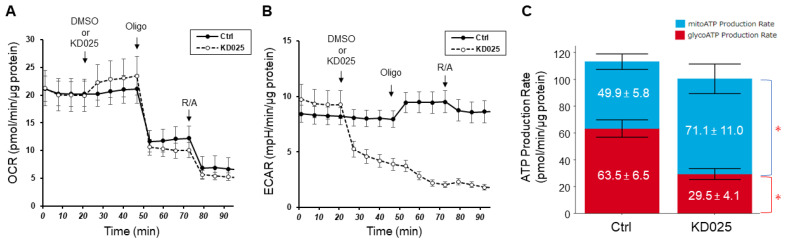
Real-time ATP rate assay after addition of 10 μM KD025 in HTM cells. OCR values (**A**) and ECAR values (**B**) in planar HTM cells. Solid lines and black circles indicate the control group that received DMSO from the injection port A (*n* = 6) and dotted lines and open circles indicate the group that received KD025 (final concentration: 10 μM) from the injection port A (*n* = 6). Calculated mitochondrial ATP production rate (blue color) and glycolytic ATP production rate (red color) using online Seahorse Analytics (https://seahorseanalytics.agilent.com) are indicated (**C**). Data are presented as means ± the standard error of the mean (SEM). * *p* < 0.05 (Student’s *t*-test).

**Figure 2 biomedicines-12-01165-f002:**
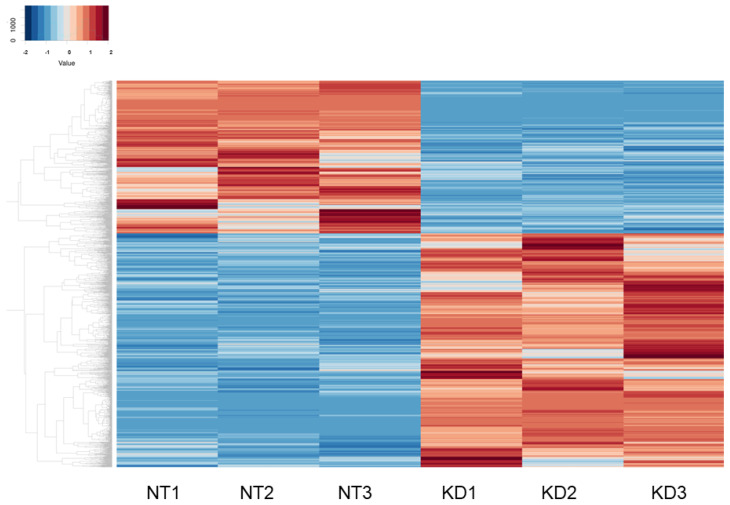
Heatmaps for DEGs for HTM cells not treated with KD025 (NT 1–3) and HTM cells treated with KD025 (KD 1–3). Differentially expressed genes (DEGs) in HTM cells not treated with KD025 (NT1~3) and HTM cells treated with KD025 (KD1~3) are shown by a hierarchical clustering heatmap. Colored points represent differentially expressed genes (cutoff FDR < 0.05) and/or the magnitude of change ≥2 that are either overexpressed (red) or underexpressed (blue) in KD HTM cells compared with their expression levels in NT HTM cells.

**Figure 3 biomedicines-12-01165-f003:**
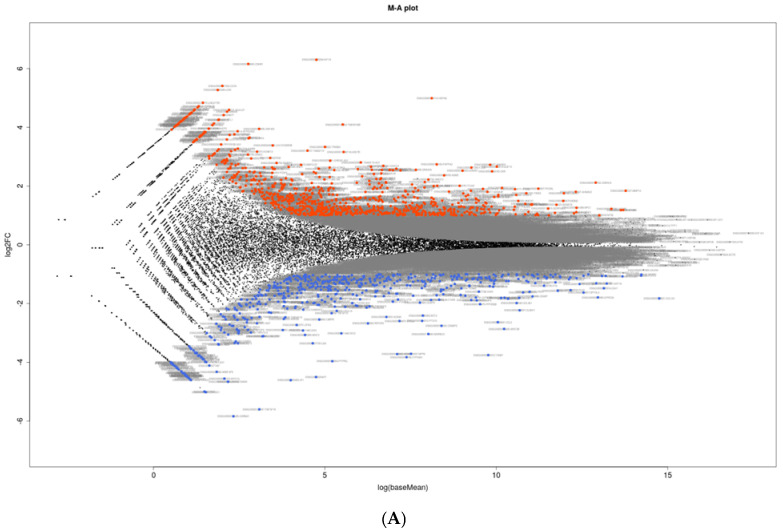
M-A (**A**) and volcano (**B**) plots for untreated HTM cells vs. KD025-treated HTM cells. Differentially expressed genes (DEGs) in HTM cells not treated with KD025 (NT) and HTM cells treated with KD025 (KD) are shown by an M–A (**A**) and a volcano (**B**) plot. Red and blue points represent overexpressed and underexpressed in KD HTM cells in comparison with their expression levels in NT (non-treated control) HTM cells.

**Figure 4 biomedicines-12-01165-f004:**
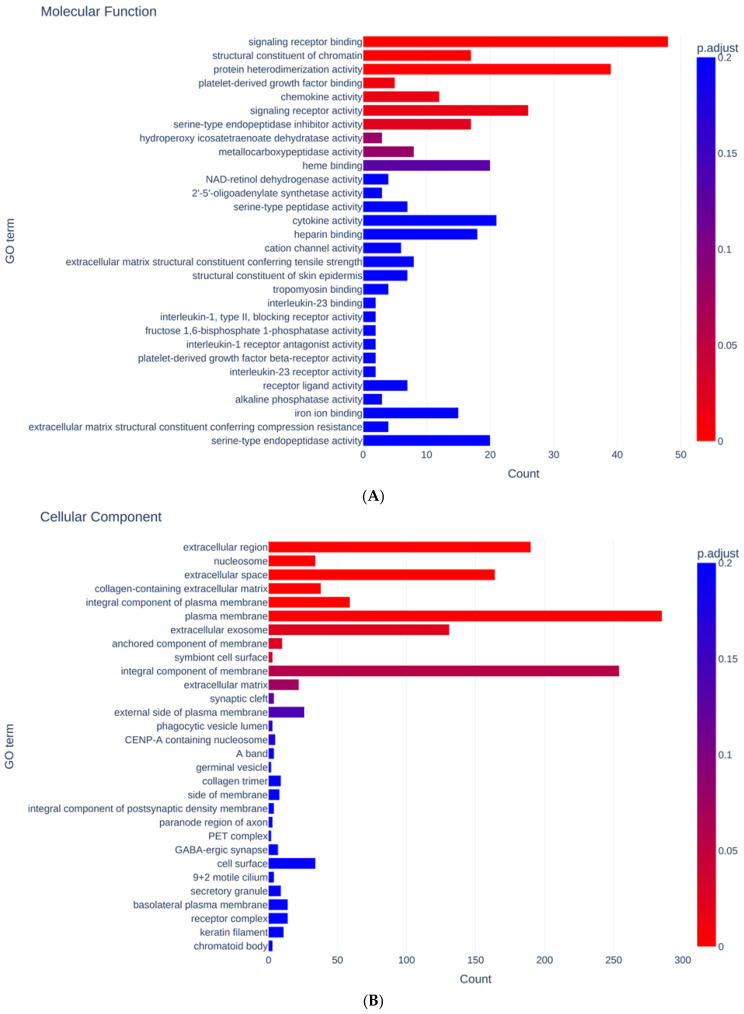
Results in GO enrichment analysis. Bar charts showing the GO terms for cellular component (**A**) and molecular function (**B**). Bar color represent *p* adjusted values (<0.2) and x-axis represents numbers of DEGs.

**Figure 5 biomedicines-12-01165-f005:**
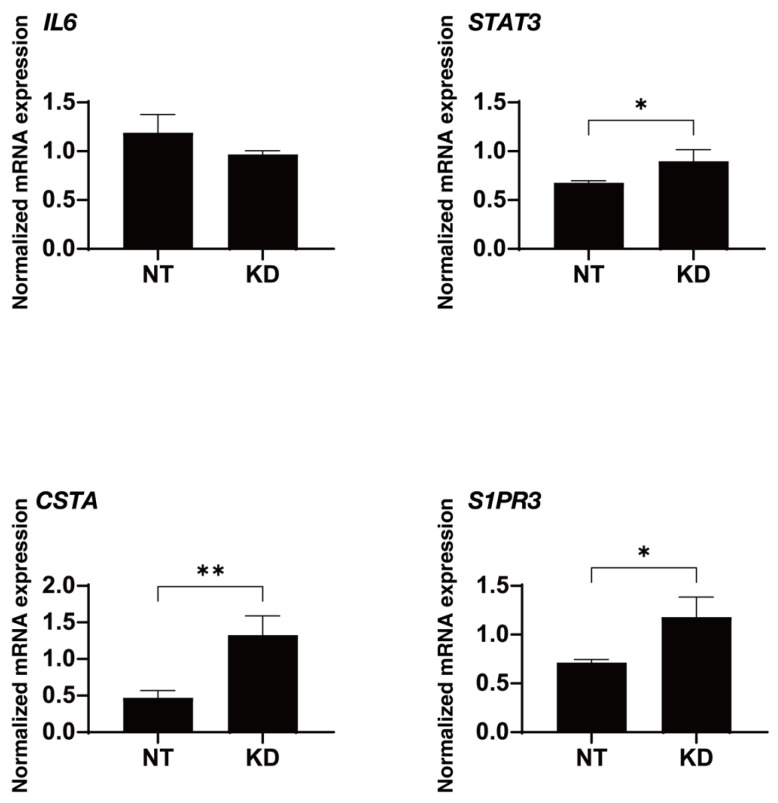
mRNA expression of STAT3, IL6, S1PR3 and CSTA in the absence and presence of 10 μM KD025. qPCR analysis of *STAT3*, *IL6*, *S1PR3* and *CSTA* of 10 μM KD025-treated and untreated 2D HTM cells was performed in duplicate, and the expression ratio as compared with reference gene, 36B4 were plotted. * *p* < 0.05, ** *p* < 0.01.

**Table 1 biomedicines-12-01165-t001:** Top 10 DEGs up-regulated and down-regulated in HTM cells not treated with KD025 and HTM cells treated with KD025.

Up Regulation		Down Regulation	
Molecules	*p*-Value	Molecules	*p*-Value
KIF1A	6.301	CHRNA1	−5.838
CDHR1	6.156	TNFSF18	−5.603
CSTA	5.410	IDO1	−5.018
LCN1	5.271	LOC105377155	−4.997
HSPA6	4.990	LOC102724434	−4.661
LINC01705	4.837	ENSG00000289492	−4.658
PIP5K1B	4.719	CFI	−4.610
LINC02568	4.679	AKAP13-AS1	−4.605
ZBBX	4.607	MYO16	−4.555
AGA-DT	4.593	RP11_885L141	−4.502

**Table 2 biomedicines-12-01165-t002:** TOP molecular and cellular functions.

Molecular and Cellular Functions		
Name	*p*-Value of Range	Molecules
Cellular Movement	1.58 × 10^−0.6^–1.98 × 10^−21^	278
Cellular Development	1.47 × 10^−0.6^–4.12 × 10^−19^	368
Cellular Function and Maintenance	5.00 × 10^−0.7^–4.12× 10^−19^	303
Cellular Growth and Proliferation	1.24 × 10^−0.6^–4.12 × 10^−19^	359
Cell-To-Cell Signaling and Interaction	1.58 × 10^−0.6^–6.61 × 10^−16^	227

**Table 3 biomedicines-12-01165-t003:** Top networks. A list of the top significant five networks with the IPA score more than 30.

Top Networks		
No.	Associated Network Functions	Score
1	Developmental Disorder, Hereditary Disorder, Organismal Injury and Abnormalities	41
2	Cardiac Dilation, Cardiac Enlargement, Cardiovascular Disease	33
3	Nervous System Development and Function, Neurological Disease, Organismal Injury and Abnormalities	33
4	Organ Morphology, Organismal Development, Renal and Urological System Development and Function	33
5	Cell Death and Survival, Organismal Injury and Abnormalities, Skeletal and Muscular Disorders	33

**Table 4 biomedicines-12-01165-t004:** Upstream regulator and causal network regulator.

Upstream Regulator	Expression Log Ratio	Activation z-Score	*p* Value of Overlap
IL6	−1.688	−2.474	0.00002
STAT3	1.51	3.061	0.00008
**Causal Network Regulator**	**Expression Log Ratio**	**Activation z-Score**	***p* Value of Overlap**
CSTA	5.41	3.312	0.001
S1PR3	1.52	4.427	0.002

**Table 5 biomedicines-12-01165-t005:** Summary of effects of Rip or KD025 on physical properties and cellular metabolic functions of dexamethasone (D)-treated and TGF-b2 (T)-treated HTM cells.

	D + Rip	D + KD	T + Rip	T + KD
TEER	↑	↑↑↑	→	↓↓↓
Size	↑↑↑↑	↓	↑↑↑↑	↑
Stiffness	↑↑↑	↓↓↓	↓↓↓	↑↑↑
OCR	→	↑	↓	↑↑
ECRA	↑	↓	↓	↓↓

## Data Availability

Experimental data will be available in case of request to corresponding author.

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
