# Peer review of "The Specific ROCK2 Inhibitor KD025 Alleviates Glycolysis through Modulating STAT3-, CSTA- and S1PR3-Linked Signaling in Human Trabecular Meshwork Cells"

_biomedicines, 2024, doi:10.3390/biomedicines12061165_

Round 1
Reviewer 1 Report
Comments and Suggestions for Authors
The manuscript of “The specific ROCK2 inhibitor KD025 alleviates glycolysis through suppressing IL-6/STAT3 signaling axis in human trabecular meshwork cells” by Megumi Watanabe and co-authors aims to study the role of Rho-associated coiled-coil-containing protein kinase 2 (ROCK2) in the human trabecular meshwork (HTM). The authors used an inhibitory analysis using the specific ROCK2 inhibitor KD025 to investigate changes in metabolic phenotype and gene expression patterns in planar cultured HTM cells. Using the Seahorse XFe96 Bioanalyzer, they found that KD025 suppressed glycolytic ATP production rate and increased mitochondrial ATP production rate in these cells. Based on RNA sequencing results followed by analysis of upstream regulators and causal networks, the authors concluded that ROCK2 mediates the IL-6/STAT3 signaling axis associated with essential metabolic pathways such as glycolysis in HTM cells.
The relevance of the research topic is due to the sharp increase in the number of ocular diseases throughout the world and the urgent need to find new molecular targets for their effective treatment. It is well known that the trabecular meshwork is responsible for most of the outflow of aqueous humor, and its disorders are associated with most cases of glaucoma when the outflow is reduced. The regulators of remodeling of the actin cytoskeleton ROCKs are involved in the ocular physiology and pathology of various ocular diseases and may be promising therapeutic targets for these diseases, especially glaucoma. According to the World Glaucoma Association (WGA), glaucoma, which is the leading cause of blindness, affects more than 80 million people worldwide, and this number is expected to increase to more than 111 million by 2040.
The manuscript is well written, covers a large body of literature (62 sources), and contributes to the understanding of the involvement of ROCK2 in signal transduction mechanisms in HTM cells.
Minor comments:
1. It is established that belumosudil (KD025) is a selective, ATP-competitive inhibitor of human ROCK2 (IC50 value of 105 nM) with some effects on human ROCK1 (IC50 value of 24 µM). The authors should explain their choice of the drug dosage (10 µM) used in the experiments.
2. Figure 5 (and Table 4) contains redundant line number labels, making the y-axis names difficult to read.
3. Instead of “Normalized mRNA expression” label in Figure 5, it would be better to indicate the name of the specific gene for which normalization was carried out. This information is also missing from the Materials and Methods section. The number of independent experiments should be included.
Comments on the Quality of English LanguageMinor editing of English language is required.
Author Response
Dear Editor,
Thank you very much for the constructive comments concerning our manuscript “The specific ROCK2 inhibitor KD025 alleviates glycolysis through suppressing IL-6/STAT3 signaling axis in human trabecular meshwork cells”. We carefully checked all of the Editor and Reviewer comments and prepared a revised version of our paper that takes these comments into account. The changes are listed below.
Editor comment
As pointed out, duplication with previous publication was reduced and changes are highlighted with blue. In addition, I apologize that labeling of Fig. 5 (NT and KD were reversed with each other) was wrong and thus corrected.
Reviewer comments:
Reviewer 1
The manuscript of “The specific ROCK2 inhibitor KD025 alleviates glycolysis through suppressing IL-6/STAT3 signaling axis in human trabecular meshwork cells” by Megumi Watanabe and co-authors aims to study the role of Rho-associated coiled-coil-containing protein kinase 2 (ROCK2) in the human trabecular meshwork (HTM). The authors used an inhibitory analysis using the specific ROCK2 inhibitor KD025 to investigate changes in metabolic phenotype and gene expression patterns in planar cultured HTM cells. Using the Seahorse XFe96 Bioanalyzer, they found that KD025 suppressed glycolytic ATP production rate and increased mitochondrial ATP production rate in these cells. Based on RNA sequencing results followed by analysis of upstream regulators and causal networks, the authors concluded that ROCK2 mediates the IL-6/STAT3 signaling axis associated with essential metabolic pathways such as glycolysis in HTM cells.
The relevance of the research topic is due to the sharp increase in the number of ocular diseases throughout the world and the urgent need to find new molecular targets for their effective treatment. It is well known that the trabecular meshwork is responsible for most of the outflow of aqueous humor, and its disorders are associated with most cases of glaucoma when the outflow is reduced. The regulators of remodeling of the actin cytoskeleton ROCKs are involved in the ocular physiology and pathology of various ocular diseases and may be promising therapeutic targets for these diseases, especially glaucoma. According to the World Glaucoma Association (WGA), glaucoma, which is the leading cause of blindness, affects more than 80 million people worldwide, and this number is expected to increase to more than 111 million by 2040.
The manuscript is well written, covers a large body of literature (62 sources), and contributes to the understanding of the involvement of ROCK2 in signal transduction mechanisms in HTM cells.
Minor comments:
- It is established that belumosudil (KD025) is a selective, ATP-competitive inhibitor of human ROCK2 (IC50 value of 105 nM) with some effects on human ROCK1 (IC50 value of 24 µM). The authors should explain their choice of the drug dosage (10 µM) used in the experiments.
Answer; Thank you for this critical comment. As for drug dosages of KD 025, previous studies using cultured cells used following concentration ranges; 7.5 mM (endothelial cell, PMID: 32628869), 3-5 mM (adipocyte, PMID: 30860936, PMID: 34443331) and 2-20 mM (brain endothelial cell, PMID: 28510599). In addition, our previous study, using 10 mM KD025 that is almost similar concentration range, significant difference in biological activities of HTM cell (PMID: 34770791, PMID: 34315994, PMID: 35740359), corneal stromal cell (PMID: 35523828), HconF cells (PMID: 34298955), orbital fibroblast (PMID: 34632320) and 3T3-L1 cells (PMID: 35877378) as compared with panROCK inhibitors. As pointed, previous study showed the selectivity for ROCK 1 and 2 inhibition (PMID: 18832915).
Based on this data, the concentration ranges (2-20 mM) in our and other studies caused less than 10% activity of ROCK2 and approximately 70-80% activity of ROCK1. Therefore, collectively, our used 10 mM concentration induces strong inhibition of ROCK2 as compared to a little inhibition of ROCK1. This information is included in the study limitation of Discussion; “Furthermore, since KD025 is a selective, ATP-competitive inhibitor of human ROCK2 (IC50 value of 105 nM) with some effects on human ROCK1 (IC50 value of 24 µM) [67], some little inhibitory effects against ROCK1 (approximately 20-30%) in contrast to 90% inhibitory effects against ROCK2 may be non-negligible in the case of present experimental condition of 10 mM KD025. Therefore, additional investigations using specific inhibitors for downstream factors related to STAT3/IL6 and other candidates as well as another ROCK2 specific inhibitors will be carried out as our next research projects.”.
- Figure 5 (and Table 4) contains redundant line number labels, making the y-axis names difficult to read.
Answer; Thank you for this comment. As pointed out, to avoid overlay of line numbers on y-axis of the Fig. 5 and Table 4, the corresponding layout was corrected.
- Instead of “Normalized mRNA expression” label in Figure 5, it would be better to indicate the name of the specific gene for which normalization was carried out. This information is also missing from the Materials and Methods section. The number of independent experiments should be included.
Answer; Thank you for comment. As suggested, the name of the specific gene for which normalization is included “Total RNA extraction, reverse transcription and real-time PCR were carried out essentially as previously reported [32,33] using predesigned primers and probes (supplemental Table 1). The expression of each respective gene was normalized by using the expression of a housekeeping gene 36B4 (Rplp0).”.
Reviewer 2
The study by Megumi Watanabe et al. is dedicated to investigating the consequences of inhibition of ROCK2 inhibition by KD025 in HTM cells. ROCK2 is an important regulator of cytoskeletal remodeling, differentiation, apoptosis, glucose metabolism and inflammation, which is involved in the pathogenesis of various eye diseases, including glaucoma. In this study, numerous shifts in gene expression patterns were observed after treatment of glaucomatous HTM cells with KD025, making this substance a potential drug for glaucoma treatment. Despite the importance of the topic investigated and the results obtained, the study has some serious shortcomings:
- In the experiments with SeaHorse, the time of incubation of the cells with KD025 and its concentration can be easily read from the graph. However, the time of incubation of the HTM cells with KD025 before obtaining samples for RNAseq and the concentration are not indicated. Please provide this information in the Materials and Methods section;
Answer; Thank you for this comment. As pointed out, incubation period was missing. For RNA sequencing analysis, 10 mM KD025 was exposed for 24 hours and this information is included.
- After performing RNAseq, the authors further analyzed some potential upstream regulators using IPA analysis. Please describe in more detail why only the expression of IL6, STAT3, CSTA and S1PR3 was selected for estimation by qPCR. Please provide the data on the change in expression of these genes in HTM cells treated with KD025 from RNAseq;
Answer; Thank you for this comment. In terms of upstream regulators and causal network regulators estimation by IPA analysis based on detected DEGs, we observed more molecules and factors other than IL6, STAT3, CSTA and S1PR3. However, most of those were chemical drugs, chemical reagents, chemical toxicants, peptides in addition to few cytokines, growth factors, kinases and transcription regulators. Among latter molecules, IL6, STAT3, CSTA and S1PR3 were most significantly changed (expression log ratio more than 1.5, IL6; 1.688, STAT3; -1.51, CSTA; -1.6 and S1PR3; -1.52) and therefore we selected these four factors. This information is included in the result section; “To study further, we estimated what kinds of upstream regulatory mechanisms were involved. IPA analysis suggested various candidates of molecules and factors as the possible upstream and causal network regulators. However, most of those were chemical drugs, chemical reagents, chemical toxicants, peptides in addition to few cytokines, growth factors, kinases and transcription regulators. Among latter molecules, interleukin 6 (IL6) and signal transducer and activator of transcription 3 (STAT3), and cystatin A (CSTA) and sphingosine-1-phosphate (S1P) receptor 3 (S1PR3) were determined as possible upstream regulators and that was a possible causal network regulator, respectively, because changes of expression rates of those four factors (expression log ratio more than |1.5|, IL6; -1.688, STAT3; 1.51, CSTA; 5.41 and S1PR3; 1.52) were much larger than those of others.”, in addition to include new Table 4.
- the authors should explain the methods and results of RNAseq in more detail, with all the statistical significance for each type and each step of the analysis. The analysis of canonical pathways is not sufficiently described. Please also indicate whether the data in Table 3 are significant;
Answer; Thank you for this comment. As suggested, additional information related to each type analysis was added; 1) GO ontology, Figure 3 legend was changed to “Bar charts showing the GO terms for cellular component (A) and molecular function (B). Bar color represent p adjusted values (<0.2) and x-axis represents numbers of DEGs.”, 2) top molecular and cellular function, P-value of range was already indicated, 3) top network, table legend is included; “A list of the top significant five networks with the IPA score more than 30.”.
- Could the authors explain why they used FDR and q values in the RNAseq analysis (line 145)? For this type of differential expression analysis, it is sufficient to use a significance criterion;
Answer; Thank you for this comment. As determination for DEGs, several criteria have been available. In the present study, to strictly determine DEGs between groups, FDR and q value were used as followed by previous studies (PMID: 26474785, PMID: 37940862).
- The authors have stated in line 144 that they used an empirical analysis. Could the authors please explain why they used this analysis and for what purposes?
Answer; Thank you for this comment. I was misunderstood that DEGs were determined just by fold-change ≧ 2.0 and false discovery rate (FDR)-adjusted P-value < 0.05 and q < 0.08, but by empirical analysis in this study. Therefore this sentence was changed; “Differentially expressed genes (DEGs) were determined as genes with fold-change ≧ 2.0 and false discovery rate (FDR)-adjusted P-value < 0.05 and q < 0.08 between groups.”.
- Table 4 shows that Rip and KD025 have different effects on HTM cells. In the Discussion section of the manuscript, please explain whether the changes in expression profile after treatment with KD025 are similar to other ROCK2 inhibitors?
Answer; Thank you for this comment. I appreciate this critical comment, and I totally agree that it would be better to test other ROCK2 inhibitors. Nevertheless, I did not test effects of other ROCK2 inhibitors except KD025. Therefore, this information if included in the study limitation in the Discussion; “Furthermore, since KD025 is a selective, ATP-competitive inhibitor of human ROCK2 (IC50 value of 105 nM) with some effects on human ROCK1 (IC50 value of 24 µM) [67], some little inhibitory effects against ROCK1 (approximately 20-30%) in contrast to 90% inhibitory effects against ROCK2 may be non-negligible in the case of present experimental condition of 10 mM KD025. Therefore, additional investigations using specific inhibitors for downstream factors related to STAT3/IL6 and other candidates as well as another ROCK2 specific inhibitors will be carried out as our next research projects.”.
- Please provide the bioinformatics analysis diagrams in better resolution: the labels and symbols in the figures are now difficult to read. It would also be helpful to the readers if the authors would add the names of the genes on the heatmap in Figure 2 and enlarge the names on the axes and legend in Figure 3, as well as add the references where are graphs A and B are located;
Answer; Thank you for this comment. As suggested, for better understanding these data, details of gene name on y-axis, M-A plot and volcano-plot were included in new supplemental materials.
- I recommend that the authors add the Conclusion at the end of the Discussion section;
Answer; Thank you for this comment. As suggested, the conclusion is included at the end of Discussion; “In conclusion, we found that a specific ROCK2 inhibitor KD025 significantly suppressed glycolytic ATP production rate and increased mitochondrial ATP production rate in HTM cells. In addition, RNA sequencing analysis estimated that 602 up-regulated and 380 down-regulated DEGs between KD025 untreated and treated HTM cells. Based on IPA analysis of these DEGs, it was estimated that ROCK2 inhibition by KD025 modulated IL-6/STAT3, CSTA and S1PR3 linked signaling related to basic biological activities such as glycolysis in HTM cells.”.
Reviewer 3
This paper entitled “The specific ROCK2 inhibitor KD025 alleviates glycolysis 2 through suppressing IL-6/STAT3 signaling axis in human trabecular meshwork cells”, provides some potentially interesting data, but unfortunately results are too preliminary to warrant publication at this time.
Specific concerns
- Rationale for looking at metabolic effects is not apparent.
Answer; Thank you very much for pointing this out. Cellular metabolism is closely related to cellular function. While there is the fact that ROCK inhibitors are promising therapeutic agents for glaucoma, their molecular mechanisms and the role of ROCK1 and ROCK2 have not been fully elucidated. Therefore, the aim of this study was to elucidate the comprehensive biological role of KD025, a specific inhibitor of ROCK2, on cellular metabolism and cellular function in trabecular meshwork using HTM cells. Interestingly, treatment with KD025 markedly suppressed ATP production by glycolysis while enhancing mitochondrial ATP-producing capacity. Evaluation of these metabolic phenotypes may be related to barrier functions such as cellular rigidity. To the best of our knowledge, both the present experimental protocol and the findings have not been reported previously, and we believe that the present finding will open the way for further researches on glaucoma and ROCK-associated signaling.
- There is no evidence ruling out the possibility of indirect effects by KD025 on metabolism.
Answer; We sincerely appreciate this careful evaluation. While it is well known that KD025 is highly selective for ROCK2 inhibition, we acknowledge that we cannot rule out the possibility that KD025 has indirect effects because the present study did not use genetic modifications or multiple inhibitors for multiple ROCK2-related signaling, as the reviewer pointed out. However, numerous additional experiments and efforts with consuming time are required to prove that KD025 does not have indirect effects rather than ROCK2 inhibition. To the best of our knowledge, the results of the present experiments have not been previously reported ever. Thus, we have decided to mention this limitation of the possibility that the effect of KD025 obtained in the present results is not mediated by ROCK2 inhibition in the revised Discussion section. We sincerely hope that the reviewer will understand the novelty of this study and our direction.
- There is only one figure on metabolic effects, and without further experimentation it is not possible to draw any conclusions from these limited data.
Answer; Thank you very much for your careful evaluation. In relation to the above comments, we acknowledge that the present study is a comprehensive results-based descriptive study rather than a mechanistic study. Thus, we have clearly stated this limitation in the revised manuscript.
- The RNA sequencing analysis is a lot of data that doesn’t say much. There were “602 significantly up-regulated and 380 significantly down-regulated differentially expressed genes.” Almost any conclusion is possible given that amount of change.
Answer; Thank you so much for this comment. Evidently, using these 602 significantly up-regulated and 380 significantly down-regulated DEGs, IPA analysis conducted various analysis including molecular and cellular functions analysis, causal network analysis, up-stream regulator and causal network regulator analysis. However, as pointed out, such information was insufficiently described. Therefore, first sentence of 3rd paragraph of Result was corrected; “Next, we conducted Ingenuity Pathway Analysis (IPA, Qiagen, Redwood City, CA) to estimate that what kinds of biological functions and networks are underlaid in the KD025 induced effects on HTM cells based upon 602 significantly up-regulated and 380 significantly down-regulated DEGs.”.
- There is not much support for the contention that interleukin 6 (IL6) is the target, certainly not enough to justify the title.
Answer; Thank you for this comment. As suggested, title was changed to “The specific ROCK2 inhibitor KD025 alleviates glycolysis through modulating STAT3, CSTA and S1PR3 related signaling in human trabecular meshwork cells”.
Other comments
- 66: Upon administering… Sentence is 6 lines, way too long and confusing.
Answer; Thank you for this comment. As pointed out, corresponding sentence was corrected to “Upon administering transforming growth factor-b2 (TGF-β2) or dexamethasone (DEX) to 2D and 3D cultured HTM cells, we observed following effects; 1) significantly smaller and stiffer 3D HTM spheroids, 2) a substantial increase in the trans-epithelial electrical resistance (TEER) values, and 3) those changes were different between TGF-b2 and DEX treatments. Based on these results, we suggested that those models may become two different in vitro glaucomatous HTM models, that is, an in vitro primary open angle glaucoma (POAG) HTM model and a steroid-induced glaucoma (SG) HTM models, respectively [22].”.
- 80: What is the difference between a ROCK inhibitor and an antagonist?
Answer; Thank you for this comment. As pointed out, inhibitor should be better and therefore, corresponding words were unified as “inhibitor” but not “antagonist”.
- 107: Cells were incubated in a CO2 free incubator, but were cultured in DMEM during this time. Did this media contain NaHCO3, which would cause a rapid rise in media pH in the absence of CO2. Need to clarify.
Answer; Thank you very much for your careful suggestion. DMEM Seahorse assay medium (#103575-100, also available at https://www.agilent.com/cs/library/datasheets/public/103575-503_Seahorse_DMEM_Medium_data%20sheet.pdf) does not contain NaHCO3, but is formulated with 5 mM HEPES, to stabilize pH in the assay medium.
- 111: How much DMSO?
Answer; Thank you for this comment. DMSO was used as a vehicle control in the present study. Our stock solution of KD025 is 10 mM dissolved in DMSO. To achieve a final concentration of 10 μM of KD025, we used a 1000-fold dilution of KD025 stock solution or DMSO. Thus, 1:1000 dilution of DMSO in Seahorse assay medium was applied in the present study.
- 126: missing micromolar
Answer; Thank you for this comment. As pointed out, this wrong font was corrected.
- 171: missing micromolar
Answer; Thank you for this comment. As pointed out, this wrong font was corrected.
- 179: How long were the cells treated for the RNA analysis?
Answer; Thank you for this comment. As pointed out, incubation period was missing. For RNA sequencing analysis, KD025 was exposed for 24 hours and this information is included.

Reviewer 2 Report
Comments and Suggestions for Authors
The study by Megumi Watanabe et al. is dedicated to investigating the consequences of inhibition of ROCK2 inhibition by KD025 in HTM cells. ROCK2 is an important regulator of cytoskeletal remodeling, differentiation, apoptosis, glucose metabolism and inflammation, which is involved in the pathogenesis of various eye diseases, including glaucoma. In this study, numerous shifts in gene expression patterns were observed after treatment of glaucomatous HTM cells with KD025, making this substance a potential drug for glaucoma treatment. Despite the importance of the topic investigated and the results obtained, the study has some serious shortcomings:
1. In the experiments with SeaHorse, the time of incubation of the cells with KD025 and its concentration can be easily read from the graph. However, the time of incubation of the HTM cells with KD025 before obtaining samples for RNAseq and the concentration are not indicated. Please provide this information in the Materials and Methods section;
2. After performing RNAseq, the authors further analyzed some potential upstream regulators using IPA analysis. Please describe in more detail why only the expression of IL6, STAT3, CSTA and S1PR3 was selected for estimation by qPCR. Please provide the data on the change in expression of these genes in HTM cells treated with KD025 from RNAseq;
3. The authors should explain the methods and results of RNAseq in more detail, with all the statistical significance for each type and each step of the analysis. The analysis of canonical pathways is not sufficiently described. Please also indicate whether the data in Table 3 are significant;
4. Could the authors explain why they used FDR and q values in the RNAseq analysis (line 145)? For this type of differential expression analysis, it is sufficient to use a significance criterion;
5. The authors have stated in line 144 that they used an empirical analysis. Could the authors please explain why they used this analysis and for what purposes?
6. Table 4 shows that Rip and KD025 have different effects on HTM cells. In the Discussion section of the manuscript, please explain whether the changes in expression profile after treatment with KD025 are similar to other ROCK2 inhibitors?
7. Please provide the bioinformatics analysis diagrams in better resolution: the labels and symbols in the figures are now difficult to read. It would also be helpful to the readers if the authors would add the names of the genes on the heatmap in Figure 2 and enlarge the names on the axes and legend in Figure 3, as well as add the references where are graphs A and B are located;
8. I recommend that the authors add the Conclusion at the end of the Discussion section;
Comments on the Quality of English LanguageEnglish Language is fine and readable, no specific comments
Author Response

(The authors gave the same response as above.)

Reviewer 3 Report
Comments and Suggestions for Authors
This paper entitled “The specific ROCK2 inhibitor KD025 alleviates glycolysis 2 through suppressing IL-6/STAT3 signaling axis in human trabecular meshwork cells”, provides some potentially interesting data, but unfortunately results are too preliminary to warrant publication at this time.
Specific concerns
1) Rationale for looking at metabolic effects is not apparent.
2) There is no evidence ruling out the possibility of indirect effects by KD025 on metabolism.
3) There is only one figure on metabolic effects, and without further experimentation it is not possible to draw any conclusions from these limited data.
4) The RNA sequencing analysis is a lot of data that doesn’t say much. There were “602 significantly up-regulated and 380 significantly down-regulated differentially expressed genes.” Almost any conclusion is possible given that amount of change.
5) There is not much support for the contention that interleukin 6 (IL6) is the target, certainly not enough to justify the title.
Other comments
66: Upon administering… Sentence is 6 lines, way too long and confusing.
80: What is the difference between a ROCK inhibitor and an antagonist?
107: Cells were incubated in a CO2 free incubator, but were cultured in DMEM during this time. Did this media contain NaHCO3, which would cause a rapid rise in media pH in the absence of CO2. Need to clarify.
111: How much DMSO?
126: missing micromolar
171: missing micromolar
179: How long were the cells treated for the RNA analysis?
Comments on the Quality of English LanguageSome minor issues.
Author Response

(The authors gave the same response as above.)

Round 2
Reviewer 2 Report
Comments and Suggestions for Authors
The authors have answered all the questions and corrected the manuscript according to the reviewer's comments. The bioinformatic analysis has been significantly improved and gives no cause for complaint. I have no additional comments
Reviewer 3 Report
Comments and Suggestions for Authors
No comments.